# Crop Production and Phosphorus Legacy with Long-Term Phosphorus- and Nitrogen-Based Swine Manure Applications under Corn-Soybean Rotation

Yan Zhang [1,2,3] 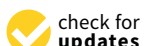, Tiequan Zhang [2,*], Yutao Wang [2], Chinsheng Tan [2], Lei Zhang [1,4], Xinhua He [5], Tom Welacky [2], Xiulan Che [3], Xiaodong Tang [3] and Zhengyin Wang [1,*]

1 College of Resources and Environment, Southwest University, Chongqing 400716, China; clock6767zhangyan@aliyun.com (Y.Z.); echozhanglei@swu.edu.cn (L.Z.)
2 Harrow Research and Development Center, Agriculture and Agri-Food Canada, Harrow, ON N0R 1G0, Canada; yutao.wang@canada.ca (Y.W.); ctan2@cogeco.ca (C.T.); gosoytom@cogeco.ca (T.W.)
3 Agricultural and Rural Committee of District Changshou, Chongqing 410220, China; Chexiulan425@163.com (X.C.); txd081138@163.com (X.T.)
4 National Base of International S&T Collaboration on Water Environmental Monitoring and Simulation in Three Gorges Reservoir Region, Chongqing 400716, China
5 Department of Land, Air and Water Resources, University of California at Davis, Davis, CA 95616, USA; xhhe@ucdavis.edu
* Correspondence: Tiequan.Zhang@canada.ca (T.Z.); wangzy16@swu.edu.cn (Z.W.)

**Abstract:** The traditional manure management strategy, based on crop N needs, results in accumulation of phosphorus (P) in soil due to the imbalance of N/P ratio between crop requirement and manure supply. This study was conducted from 2004 to 2013 to evaluate the effects of P-based liquid and solid swine manure (LMP and SMP, for P-based liquid and solid swine manure, respectively) application, in comparison with N-based application (LMN and SMN, for N-based liquid and solid swine manure, respectively), on crop yield and soil residual P under corn (*Zea mays* L.)–soybean (*Glycine max* L.) rotation in a Brookston clay loam soil of the Lake Erie basin, ON, Canada. Chemical fertilizer P (CFP) and non-P treatments were included as controls (CK). For liquid manure treatments, corn yield for LMN showed a lower annual corn yield (7.82 Mg ha$^{-1}$) than LMP (9.36 Mg ha$^{-1}$), and their differences were even statistically significant at $p < 0.05$ in some cropping years. The annual corn yield of LMP was also higher than those of SMP (7.45 Mg ha$^{-1}$) and SMN (7.41 Mg ha$^{-1}$), even the CFP (8.61 Mg ha$^{-1}$), although the corresponding yield differences were not significant ($p < 0.05$) in some cropping years. For soybean, the plots with P application produced an average of 0.98 Mg ha$^{-1}$ greater annual yields than CK. No significant differences were found between CFP and manure treatments. The annual corn yield of SMN was close to that of the CK (7.19 Mg ha$^{-1}$). The grain P removal (GPR) of SMN (18.6 kg ha$^{-1}$) for soybean was significantly higher than that of the other treatments. The above-ground-P uptake (AGPU) in SMN, for both corn and soybean, was significantly higher than that of the other five treatments. The soil test P (STP) presented clear stratification, concentrating in the top 30 cm soil depth after 10 years of application. The contents of STP with LMN and SMN increased from 7.1 mg P kg$^{-1}$ to 12.4 and 45.5 mg P kg$^{-1}$, respectively. The sum of STP mass (0–30 cm) with LMP (31.6 kg ha$^{-1}$) was largely identical to that with CFP (30.1 kg ha$^{-1}$); however, with SMN (173.7 kg ha$^{-1}$), it was significantly higher than the rest of the treatments. Manure P source availability coefficients were averaged at 1.06 and 1.07 for LMP and SMP, respectively. The addition of phosphorus-based liquid or solid swine manure can overcome the drawback of traditional N-based applications by potentially reducing the adverse impact on water quality while sustaining crop agronomic production.

**Keywords:** swine manure; crop yields; phosphorus-based manure application; nitrogen-based manure application; manure phosphorus source availability coefficient; phosphorus use efficiency; phosphorus availability; soil test P

## 1. Introduction

With the growing demands for animal protein, the large quantity of manure engendered in livestock husbandry has been a great threat to water ecology. Optimized application can make manure an excellent source of nutrients for crops and an economical fertilizer for producers [1]. The strategies of replacing mineral fertilizers with manure can reduce the dependency on rock phosphate, a non-renewable and geographically restricted natural resource, and improve food security [2–4]. However, phosphorus (P) forms in manures vary, with fractions of P bounding in slowly dissolving compounds [5]. A buildup of soil P, contributed from applied manure, has increasingly become a worldwide concern as potential non-point source of pollution to watercourses [6–8]. Over the past three decades, Lake Erie, the most productive part of the Laurentian Great Lakes, has experienced substantial eutrophication with harmful algal blooms and extensive hypoxic zones in the central and western basins [9,10]. Excessive P loading to the lake water body has been particularly influential [9,11]. Phosphorus in agricultural runoff is deemed to be a major contributor to algal bloom development [10], approximately 85% of the P originating from nonpoint sources is from manure and commercial fertilizer [12]. For instance, in the western Lake Erie watershed, 450,897 pigs were raised in concentrated animal feeding facilities (January 2017) [13]. Means of swine manure disposal should be managed to relieve anthropogenic stressors and P loading to Lake Erie, while optimizing crop yield.

Manure has traditionally been applied due to on its N supplying capacity to crops, which generally causes large residual P in soil due to lower N/P ratios in manures relative to those in crops [14–17]. The values of N: P ratios of solid swine manure, liquid swine manure and cattle manure range from 1.46 to 2.88, from 2.0 to 5.8, and from 2.35 to 2.8 [18–21], respectively, while in grain corn and soybean they are 6.1 and 10, respectively [22,23]. Field studies on cattle manure and compost showed that P-based application can achieve optimum crop yield and prevent nutrient P from buildup in the soil, while repeated annual or biennial applications based on crop N requirements often result in over-application of manure P and much greater soil P accumulation in 0- to 0.15 m soil layer [24,25]. A five-year study on a loamy Black Chernozem soil in Saskatchewan proved that higher proportion of P in the liquid swine manure than in cattle manure could be removed by the crop, which meant that P added in swine manure treatment matched more closely with P removal by crop than cattle manure treatment [26]. Additionally, more available P accumulated in the soil receiving cattle manure than receiving swine manure with similar N supplied. This was attributed to the decrease in P adsorption with increases in the degree of P saturation in the soil with cattle manure addition compared with the lower P addition of liquid swine manure [27,28]. An incubation study conducted to compare P availability from liquid swine and cattle feedlot manure suggested that there was no significant difference in P availability, however, liquid swine manure showed better distribution than solid cattle manure [29]. Commonly, liquid swine manure was confirmed to be as efficient as mineral fertilizers for the production of grain crops [30,31]; and P in liquid swine manure was even more plant available than fertilizer P [32]. A six year study with regard to N-based treatments of liquid swine manure under continuous corn or corn-soybean rotation showed that soil-test P in the surface 20 cm increased with annual manure applications but declined slightly for the rotation, partly due to the higher grain P removal in the rotated plots than in continuous corn plots [21]. Another study in grass lands in southern Manitoba suggested that the accumulation of P following swine manure application was restricted to the upper 15 cm layer of soil. Annual applications of liquid and solid swine manure at an N-based rate resulted in soil test P values twice as those in the corresponding P-based treatments, which may result in further P loading and potentially greater leaching [33].

Part of the P added with pig slurry accumulated in superficial soil layers [34]; the continuous application of swine manure increased the labile and moderately labile P in the soil plow layer but did not affect most of P fractions [35]. Phosphorus migration in the soil profile by vertical matrix, macropore, or by-pass flow could increase P concentration in deeper soil layers [36]. Successive applications of pig slurry were able to change the P

balance between liquid and solid phases in the surface soil layers, increase P mobility and concentration in soil solution, and could even leach to the depth of 40-cm soil layer [37]. Other literature has also demonstrated the risks associated with assessing potential P losses on the basis of P mobility in the topsoil alone [38–40].

For the typical corn-soybean rotation mode in the Lake Erie basin, will P-based management for solid and liquid swine manure lead to the decline in agricultural benefits or accumulation of soil P over a long term of application compared with the traditional N-based managements? How about the soil P distribution in the soil profile while concerns currently exist about increases in potential for P loss after a long-term swine manure management?

To holistically assess P use of a long-term crop production eco-system, all applied P pools should be taken into account [41,42]. Update by crops should not be assumed to be the only use for applied P. Cumulative P legacy in soil from consecutive long-term application can contribute to crop uptake and increase the yields of subsequent crops for several years [43–45]. In order to assess the field availability of P in various forms and types of manure and other organic amendments, a new approach has recently been developed, namely phosphorus source availability coefficient (PSAC), which accounts for both the long-term changes in soil test P (STP) and cumulative crop P uptake (CCPU) [46,47].

The objectives of the study were: (1) to evaluate the crop yield effects of long-term (i.e., 10 years) P-based, in comparison with N-based, liquid and solid pig manure application under corn–soybean in a clay loam soil of the Lake Erie basin; (2) to assess soil P downward movement by investigating changes in soil test P (STP) in soil profile; and (3) to determine the PSAC values of solid and liquid swine manures under long-term field conditions.

## 2. Materials and Methods

### 2.1. Field Site and Preparation

The experiment was carried out from 2004 to 2013 on a clay-loam soil (fine loamy, mixed, mesic, Typic Argiaquoll) in the Lake Erie basin at the Honorable Eugene F. Whelan Experimental Farm of Harrow Research and Development Centre, Agriculture and Agri-Food Canada, Woodslee, Ontario (42°13' N, 82°44' W). The climate is mild and humid, with average annual temperature of 8.7 °C and precipitation of 855 mm (Tables 1 and 2). Before this study, the soil (pH 6.2) contained 28% sand, 35% silt, 37% clay, 22.7 g kg$^{-1}$ organic C, 1.95 g kg$^{-1}$ total N, 7.1 mg kg$^{-1}$ Olsen P (classified as low level), and 131 mg kg$^{-1}$ acetate ammonia extractable K [46,47]. Corn was grown for three years prior to this experiment at the application rate of 150 kg N ha$^{-1}$ y$^{-1}$ to minimize any possible spatial variations in soil test P and other nutrients.

**Table 1.** Growing seasonal monthly precipitation (mm) on the experimental site, Woodslee, ON, Canada, from 2004 to 2013.

| Month | 2004 | 2005 | 2006 | 2007 | 2008 | 2009 | 2010 | 2011 | 2012 | 2013 |
|---|---|---|---|---|---|---|---|---|---|---|
| March | 46.3 | 8.2 | 54.2 | 69 | 68.8 | 96.8 | 36.4 | 84.8 | 63.4 | 19.6 |
| April | 47.5 | 62.2 | 57.4 | 66.8 | 36.8 | 114 | 61.2 | 128.4 | 30.4 | 106.4 |
| May | 166 | 22.4 | 104.4 | 53 | 54.4 | 45 | 107.8 | 193 | 83.4 | 47.0 |
| June | 79.4 | 22.8 | 66.8 | 58 | 186.2 | 85.6 | 113.2 | 62.6 | 16.8 | 162.8 |
| July | 89.8 | 61.6 | 108.2 | 36 | 85.6 | 63.8 | 148.4 | 120 | 102.2 | 185.9 |
| August | 124.2 | 51 | 76 | 111.2 | 12.8 | 90.6 | 9.2 | 104.8 | 109.6 | 36.6 |
| September | 20 | 66.4 | 59.6 | 62.2 | 124.6 | 20.8 | 90.4 | 180.6 | 30.8 | 107.9 |
| October | 56 | 8.6 | 108.2 | 58.6 | 28.8 | 75.8 | 63 | 112 | 58 | 64.0 |
| November | 68.8 | 60.8 | 101.4 | 61.6 | 94.8 | 17.4 | 85.2 | 179 | 11.7 | 40.1 |
| Total | 698 | 364 | 736.2 | 576.4 | 692.8 | 609.8 | 714.8 | 1165.2 | 506.3 | 521.7 |

**Table 2.** Growing seasonal monthly mean temperature (°C) on the experimental site, Woodslee, ON, Canada, from 2004 to 2013.

| Month | 2004 | 2005 | 2006 | 2007 | 2008 | 2009 | 2010 | 2011 | 2012 | 2013 | Average |
|-------|------|------|------|------|------|------|------|------|------|------|---------|
| March | 3.7 | 0.4 | 2.8 | 3.4 | 0.4 | 2.3 | 4.3 | 0.7 | 8.8 | 0.5 | 2.7 |
| April | 8.8 | 9.5 | 10.2 | 8.4 | 9.6 | 8.1 | 11 | 6.7 | 8.2 | 7.1 | 8.8 |
| May | 14.9 | 13.1 | 15.1 | 15.7 | 13 | 14.8 | 17.1 | 14.4 | 16.8 | 16 | 15.1 |
| June | 19.2 | 22.9 | 20.1 | 21.4 | 20.7 | 19.4 | 21.9 | 20.3 | 21.5 | 20.2 | 20.8 |
| July | 21.5 | 24.3 | 23.9 | 22 | 23.1 | 20.5 | 24.8 | 24.8 | 24.7 | 22.4 | 23.2 |
| August | 20 | 23.8 | 22.4 | 22.9 | 21.7 | 21.6 | 23.6 | 22.2 | 22.2 | 21.4 | 22.2 |
| September | 19.5 | 19.2 | 17.1 | 20.3 | 19 | 18.5 | 18.4 | 17.9 | 17.7 | 18.6 | 18.6 |
| October | 12.7 | 12.9 | 10.2 | 15 | 11.1 | 10.1 | 12.3 | 11.8 | 11.6 | 13.5 | 12.1 |
| November | 7.9 | 6.4 | 6 | 5 | 4.3 | 7.5 | 6.1 | 7.3 | 4.4 | 3.3 | 5.8 |

### 2.2. Manure Sampling and Determination of Chemical Composition

From two local swine farms in south-western Ontario, solid swine manure (with wheat straw as bedding material) was collected and liquid swine manure was pumped into the tanker. Two to three days before the application to the field plots, the pile of solid manure was thoroughly mixed, the composite fresh solid manure samples were collected simultaneously from multiple locations (≥15 locations for each composite sample), and liquid swine manure lagoon was agitated for collecting the samples. Moisture, total N, P, and K were determined immediately [48]. Application rates of manures were determined by considering the contents of moisture, available N and or total P and K depending the treatments.

### 2.3. Experimental Design

The experimental design was in a randomized complete block with three replications. Each plot was 9 m by 25 m in size. For the typical corn (*Zea mays* L.)-soybean (*Glycine max* L.) rotation in Lake Erie Basin, 180 kg ha$^{-1}$ available N, 40 kg P ha$^{-1}$ and 100 kg K ha$^{-1}$ were, at the time, commonly applied at the time biennially by local farmers. To ensure sufficient supply for the rotation of corn and soybean, all treatments were adjusted to the application rates of 100 kg K ha$^{-1}$ and 200 kg available N ha$^{-1}$, while P-based treatments received 50 kg P ha$^{-1}$. This was done by considering 56% and 38% of total N that were available to crops during the growing season in liquid and solid swine manures [48,49], respectively, and the balances of the designed N and K rates were made up with ammonium nitrate (NH$_4$NO$_3$) and potassium chloride (KCL). All K in manure was assumed available to crops (Table 3). The treatments of this study included a no-P control (CK), chemical fertilizer P treatment (triple superphosphate at 50 kg P ha$^{-1}$; CFP), and P- vs. N-based solid and liquid swine manure treatments.

**Table 3.** Statistical significances of the effects of P application treatment and cropping year on grain yield, grain P removal and above-ground-P uptake (grain and stover) over the 10-year (2004–2013) period in a Brookston clay loam soil, Ontario, Canada.

| Factor | Corn | | | Soybean | | |
|--------|------|------|------|---------|------|------|
| | Grain Yield | Grain P Removal | Above-Ground-P Uptake | Grain Yield | Grain P Removal | Above-Ground-P Uptake |
| Treat [§] (T) | <0.001 | <0.001 | <0.001 | <0.001 | <0.001 | <0.001 |
| Year (Y) | <0.001 | <0.001 | <0.001 | <0.001 | <0.001 | 0.002 |
| T × Y | 0.012 | 0.038 | 0.105 | 0.105 | 0.258 | 0.398 |

[§] Treat: treatment.

Chemical fertilizers and/or manures were applied every other year to the corn phase of the corn (*Zea mays* L.)–soybean (*Glycine max* L.) rotation. Manure and chemical fertilizers were broadcast prior to planting and incorporated for all treatments immediately or as soon as weather conditions permitted. Corn was normally planted between late May and early June as weather and soil conditions allowed. Soybean planting was usually conducted one-week later than corn, also depending on weather and soil conditions. The corn-soybean rotation started with corn in 2004 followed by soybean in 2005 and continued as such until 2013. Typical corn and soybean varieties were used according to the local recommendations each year, except for 2004, in which an early-maturing corn variety (cv. N29A2, Syngenta Seeds Canada) was adopted due to delayed planting owing to wet soil conditions. Corn and soybean were sown at 77,000 and 490,000 seeds ha$^{-1}$, respectively [48,49]. Herbicides and pesticides were used in accordance with local recommendations for both corn and soybean, respectively, such as AMITROL 240, whick was applied pre-plant prior to soybean planting to provide good control of tufted vetch (scientific name: vicia cracca). The plots were ploughed to 20 cm depth in the fall after harvest. Corn and soybean were harvested between late October and early November and between early and middle October, respectively.

### 2.4. Yield Determination and Soil and Plant Sampling for P Analyses

Each year at harvest time, the central eight and 20 rows of each plot for corn and soybean, respectively, were harvested using a small-sized combine harvester to get the grain yields. The content of grain moisture was measured using a grain analyzer (GAC200, Dickey-john Crop, Cornwall, ON, USA). The grain yields were reported by adjusting to a moisture content of 15% and 13% for corn and soybean, respectively.

For each plot, the above ground part of 16 plants for corn and 40 plants for soybean were randomly selected to obtain stover biomass. Grain and stover samples were dried at 55 °C and ground to pass through a 1-mm sieve. Plant samples were digested using the $H_2SO_4$–$H_2O_2$ procedure [50]. Tissue P content in the digested solution was analyzed using the molybdate-ascorbic acid colorimetry method with a flow injection auto-analyzer (QuickChem FIA+ Auto-Analyzer, 8000 Series, Lachat Instruments, Loveland, CO, USA) [51]. Grain P removal was calculated accordingly from grain yield and P concentrations in grain. Above-ground-P uptake was the sum of grain P removal and stover P uptake. Phosphorus contained in cobs was included in stover P uptake for corn as well.

Composite soil samples (2 cores made a composite soil sample) were collected from each of the 0–7.5-, 7.5–15, 15–30, 30–50, 50–70 and 70–90 cm layer depths using a hydraulic soil core sampler for each treatment plot shortly after harvest in the fall of 2013. Simultaneously, at least twelve additional cores were taken from 0–7.5, 7.5–15, and 15–30 cm depths using a 15 cm-long and 3 cm-diameter soil core sampler to minimize potential spatial variation in the surface soil layers; then, mixed the sections from the 15 cm-long soil columns with the corresponding depth of the 90 cm-long column to make one representative soil sample of each depth. The bulk density of 0–15 and 15–30 cm layers was collected and determined for each treatment plot.

Soil samples were air dried, sieved (2 mm), and extracted using the Olsen P procedure. Phosphorus concentration was analyzed using the molybdate blue method with a Lachat QuickChem FIA+ Auto-Analyzer (8000 Series, Lachat Instruments, Loveland, CO, USA) [51–53].

### 2.5. Calculation of Phosphorus Source Availability Coefficients

The PSAC was determined as the ratio of P that was available in the plots receiving manure to those in the plots receiving chemical fertilizer P over a certain period [46].

$$PSAC = (SMS + CCPU)_{manure} /(SMS + CCPU)_{CFP}$$

The available P included P residue in soil layer, as indicated by soil test P (Olsen P), and the cumulative crop P uptake (CCPU) for the specific treatment [46]. For this 10-year

study (2004–2013), P residue in soil layers was expressed by the sum of the mass of STP (SMS) which was calculated through soil test P and the soil bulk density of the plot soil layers (0–15 cm and 15–30 cm) in 2013 after harvest. The cumulative crop P uptake was the sum of the above-ground crop P uptake each year for a given plot from 2004 to 2013.

### 2.6. Statistical Analysis

All statistical analyses were conducted using the Statistical Product and Service Solutions (IBM SPSS Statistics 21.0). GLM (general linear model) univariate analysis with multiple factors was conducted on grain yield, grain P removal and above-ground-P uptake to determine treatment and cropping year effects and their interactions. One-way ANOVA was performed using the Duncan procedure to determine the significance of differences at the level of $p \leq 0.05$ when variance was significant. Appropriate regression equations were selected on the criteria of best fit. Polynomial regressions were run before performing the significance test on the correlation parameters.

## 3. Results and Discussion

### 3.1. Crop Yield

Yields of corn and soybean were significantly affected by both P application treatment and cropping year. The interaction between year and treatment was significant for corn yield, but not for soybean yield during the 10-year period from 2004 to 2013 (Table 3).

For corn grain yield, the differences among treatments were not statistically significant in 2004, 2006 and 2008. Liquid swine manure treatments showed greater yields than solid swine manure treatments in 2012 (Figure 1). Along with fertilization years, the yields for LMP in 2010 (8.6 Mg ha$^{-1}$) was the highest among all treatments. The yields for SMP in both 2010 and 2012 (5.23 and 9.38 Mg ha$^{-1}$) were statistically identical to those of CK (5.68 and 10.44 Mg ha$^{-1}$ in 2010 and 2012, respectively). However, the yield for SMN (5.65 Mg ha$^{-1}$) was close to that of CK in 2010, and exceeded CK in 2012 (11.47 Mg ha$^{-1}$ for SMN) (Figure 1). It appears that increases in microbial activities primed from the organic matter addition of solid manure might have caused competitive assimilation of N, which consequently resulted in crop N deficiency [54–56].

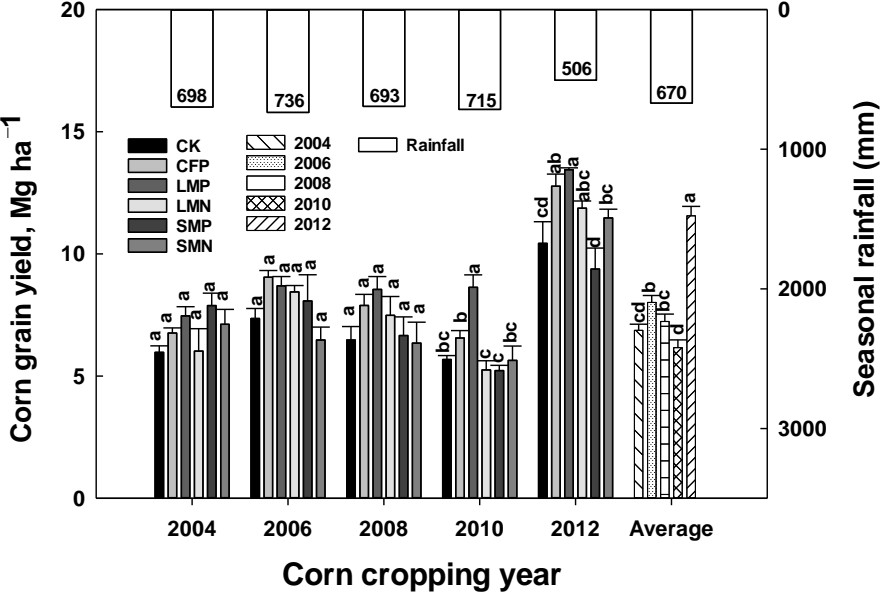

**Figure 1.** Effects of cropping year and the growing season rainfall on treatmentwise corn grain yield over 10-year (2004–2013) fertilization under corn-soybean rotation in a clay loam soil, Woodslee, ON, Canada. Error bar is the standard error of the mean (*n* = 3). Different letters over the bars indicate the yields were significantly different at the $p \leq 0.05$ level.

Applied P played an important role to improve crop yield by comparing the fertilized treatments with CK. For corn, across the five cycles of rotation, the annual yields with LMP were significantly higher than those of the CK. The annual corn yields with CFP and LMN were 1.42 and 0.63 Mg ha$^{-1}$ higher than with the CK, respectively.

For liquid manure treatments, there was a lower annual corn yield for LMN than LMP in 2010 even though the former was added at a higher P rate. The annual corn yield of LMP was significantly higher than that of SMP and SMN in 2010 and 2012, even higher than that of CFP. The highest rate of P was applied with SMN among treatments; however, the annual corn yield of SMN was close to that of the CK (7.19 Mg ha$^{-1}$). The results of annual corn yield coincided with the study conducted on a Sharpsburg silty clay loam soil, in which most of the N-based treatments showed less corn grain yield than P-based ones, and the compost treatment for N-based mode showed the lowest corn grain yield even though it was added with the highest rates of N and P among all treatments [14]. The reasons for lower corn yields with N-based applications in some cropping years need to be explicitly explained with further studies.

Amongst the cropping years, the corn yield in 2012 (11.56 Mg ha$^{-1}$) significantly exceeded those in other years, and the yield in 2010 (6.17 Mg ha$^{-1}$) was the lowest among all cropping years of corn (Figure 1).

Crop yield could be affected not only by the availability of soil nutrient but also by other factors such as climate conditions, including rainfall and temperature. A study conducted in south-east Nebraska reported that, for rainfall, corn yield under corn-soybean rotation was negatively related with the rainfall before and during planting time (late February to mid-May) and positively related with the rainfall in the anthesis and kernel-filling periods. In terms of temperature, corn yield was negatively correlated with the temperature from late May through to mid-July [57]. Another examination on corn yield in central Missouri, based on the long-term (1895–1998) datasets, characterized the high-yield years by less rainfall and warmer temperatures in the planting period, more rainfall and warmer temperatures during germination and emergence, more rainfall and cooler-than-average temperatures as the key features in the anthesis and kernel-filling periods, and less rainfall and higher temperatures in the ripening period [58].

In the current study, a similar rule regarding rainfall was obtained during February to May, as corn yield appeared to descend when the precipitation increased from 210.2 mm in 2012 to 265.4 mm in 2006 (Figure 2A). The warmer temperatures before and during planting time in 2012 might be the key factors in the highest corn yield in the year, which would agree with the previous research [58]. The average temperature in March, 2012 was 8.8 °C, which was 6.1 °C higher than the mean in March during 2004–2013. The average temperature in May, 2012 was 16.8 °C, which was 1.7 °C higher than the average temperature of May during 2004–2013 (Table 2). There were three days with the daily maximum temperature reaching up to 30 °C and three days with the daily minimum temperature at 2 °C in May, 2012. In 2010, although the average temperature in May was 17.1 °C, the lowest temperature was 0 °C for five days in May; it may be harmful to seed germination.

During the anthesis and kernel-filling periods, from July to September for corn growing in Woodslee, ON, Canada, the highest yield appeared in 2012 with the seasonal precipitation of 242.6 mm. Corn yield showed downward trends when the rainfall was less than 235 mm (e.g., 234 mm in 2004, and 223 mm in 2008) or greater than 242.6 mm (e.g., 243.8 mm in 2006 and 248 mm in 2010) (Figure 2B) from July through September. No significant relationship between the rainfall in ripening time and corn yield occurred during this 10-year experiment. However, more rainfall could reverse the effect of fertilizer N on corn growth by leading to N losses through denitrification or leaching to subsurface drainage water [1,59]. The lower rainfall during March through November in 2012 partly accounted for the remarkably higher corn yield than other corn years (Figure 2C). In addition, the sufficient pre-season precipitation in 2011 might have affected the corn yield in 2012 by

providing a potential soil water storage (Table 1), as corn yield was positively related with an index for precipitation of the 12-mo period preceding planting [60].

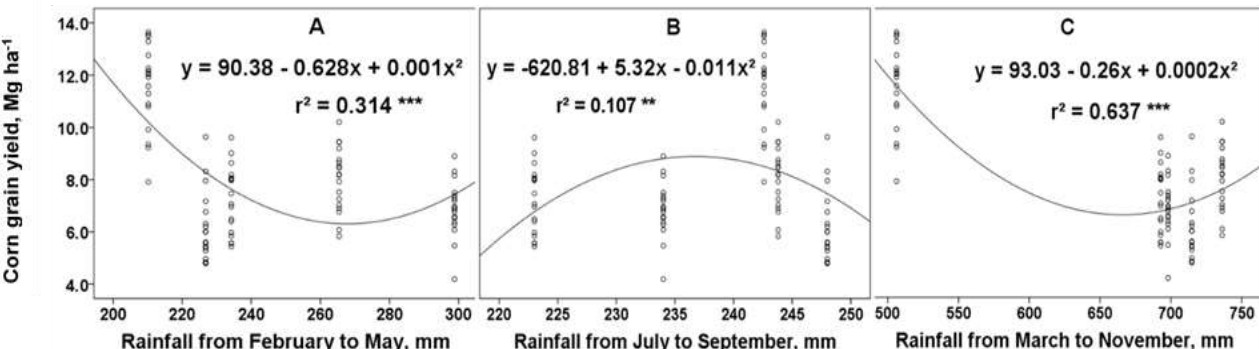

**Figure 2.** Relationships between corn yield and the rainfall for the specific periods ((**A**), February to May; (**B**), July to September; (**C**), March to November) during the seasons over 10-years (2004–2013) under corn-soybean rotation in a clay loam soil, Woodslee, ON, Canada. ** and ***, significant at $p \leq 0.01$ and 0.001, respectively.

In the anthesis and kernel-filling periods for corn in 2012, the average temperature was 1.5 °C higher than that in July, similar to the average temperature in August and 0.9 °C lower than the average temperature in September. The corn yield in 2006 (8.02 Mg ha$^{-1}$) was the second highest amongst the five corn years. The monthly average temperature was 0.7 °C and 0.2 °C higher than those in July and August, respectively, and was 1.5 °C lower than the average temperature in September. However, the average temperature in July, August and September of 2010 was higher than those in 2012 and 2006. This is partly consistent with the previous study on corn growth that cooler-than-average temperatures in the periods were characterized as one of the key features for the high-yield years [58]. Longer term of study combined with temperature, rainfall and other factors could be more appropriate in understanding how climate may affect crop yield.

Soybean plots receiving P produced significantly greater annual yields in comparison with CK. On the average, soybean plots with P application produced 0.98 Mg ha$^{-1}$ greater annual yields than CK (Figure 3). No significant differences were found between CFP and manure treatments (Figure 3). The means of soybean yield for CFP, LMP, LMN, SMP and SMN approximated to each other and averaged about 3.65 Mg ha$^{-1}$. The average yields of soybean among treatments were subtly influenced by the residual soil P despite the application differentials in forms and/or rate of P on preceding corn phase with CFP and P- and N-based approaches. Other researches also reported that the soybean yields remained near the maximum and were not affected by direct application of swine manure, nor the residual effects from the application of swine manure made in the previous year [61,62].

Significant differences of annual soybean grain yields were found among the cropping years. The yield in 2011 (3.86 Mg ha$^{-1}$) was significantly higher than those in other years except 2007. The higher yield in 2011 may have had a lot to do with the greatest rainfall relative to other soybean years (Table 2), which was unlike the previous research that the benefit of rotation for soybean grain yield did not vary with weather conditions [54]. The soybean yield was the lowest in 2005 (3.12 Mg ha$^{-1}$) among the five years.

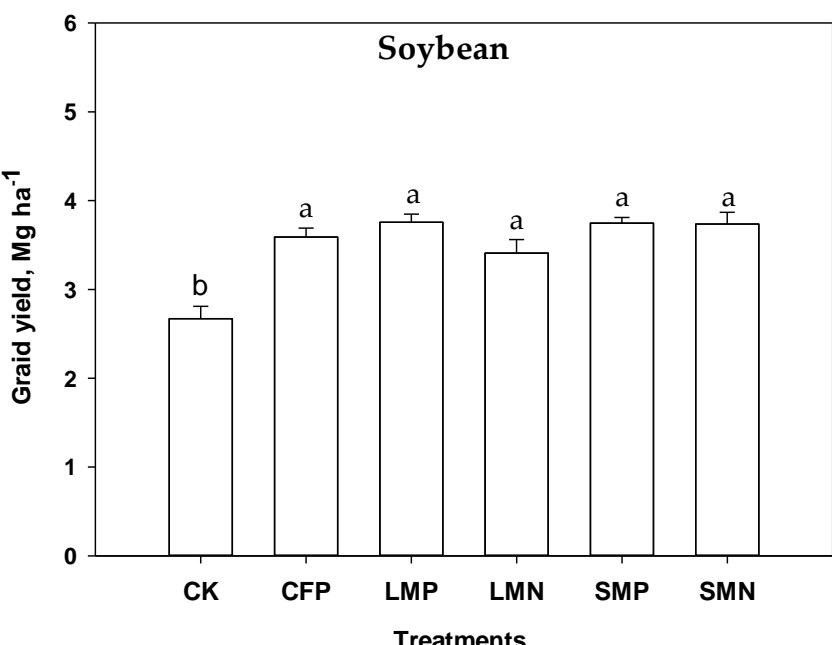

**Figure 3.** Responses of soybean grain yield to various sources of P addition (chemical fertilizer vs. liquid and solid swine manures) and application approaches (P-based vs. N-based) over a 10-year time period, 2004–2013, in a clay loam soil, Woodslee, ON, Canada. CK, no-P control; CFP, chemical fertilizer P; LMP, P-based liquid swine addition; LMN, N-based liquid swine manure addition; SMP, P-based solid swine manure addition; SMN, N-based solid swine manure addition. Error bar is the standard error of the mean ($n = 15$). The different letters over the bars indicate significant differences at the $p \leq 0.05$ level.

### 3.2. Grain P Removal and Above-Ground-P Uptake

There were significant treatment and year effects on grain P removal (GPR) and above-ground-P uptake (AGPU) of both corn and soybean and treatment by year interaction on GPR of corn (Table 3).

In the cropping years for corn, the GPR for the treatments with P addition were significantly higher compared with those of the CK in 2004, 2010 and 2012 (Figure 4). The GPR of SMN and SMP were higher than those of other treatments in 2004. The GPR of LMP was remarkably higher than that of other treatments in 2010 (Figure 4). However, across the five cycles of rotation, the average GPR amongst the treatments with P addition showed no significant difference with each other. This could be attributed from the calculation of GPR which was got from grain yield and grain P concentration (GPC). Although there were differences in grain yield among treatments, especially in 2010 and 2012, the differences might be offset when GPC was considered as another factor to obtain GPR. The following discussion in the current paper confirmed this by the correlation between GPC and P application rate.

For soybean, amongst treatments, the average GPR of the CK (10.1 kg ha$^{-1}$) appeared to be less than all the other treatments added with P; the GPR of SMN (18.6 kg ha$^{-1}$) was significantly higher than that of the other treatments. The grain P removal in 2011 (17.7 kg ha$^{-1}$) was significantly higher than that in the other cropping years except that in 2013 (15.6 kg ha$^{-1}$).

The AGPU of the treatments added with P were significantly higher than that of CK (Table 4). AGPU in SMN was significantly higher compared with those in the other five treatments for both corn and soybean. Another study carried out on cattle manure and composted amendments also showed that P removed by plant tissues was greater under N-based than under P-based amendments [24]. It seemed that the over application of P in SMN resulted in luxury consumption by the crop.

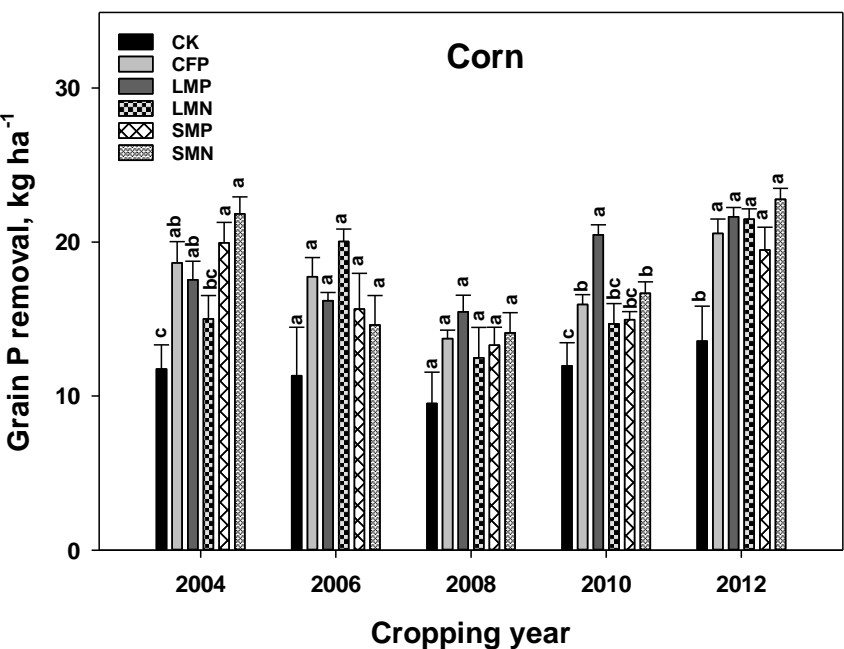

**Figure 4.** Effects of cropping year and source of P addition (chemical fertilizer vs. liquid and solid Scheme 10. year time period, 2004–2013, in a Brookston clay loam soils, Woodslee, ON, Canada. CK, no-P control; CFP, chemical fertilizer P; LMP, P-based liquid swine addition; LMN, N-based liquid swine manure addition; SMP, P-based solid swine manure addition; SMN, N-based solid swine manure addition. Error bar is the standard error of the mean (*n* = 3). The different letters over the bars indicate significant differences at the $p \leq 0.05$ level.

**Table 4.** Responses of above-ground-P uptake for corn and soybean to various sources of P addition (chemical fertilizer vs. liquid and solid swine manures) and application approaches (P-based vs. N-based) over a 10-year time period, 2004–2013, under a corn-soybean rotation in a clay loam soil, Woodslee, ON, Canada.

| | **Corn** | | **Soybean** | |
|---|---|---|---|---|
| | **Factor** | **Above-Ground-P Uptake, kg ha$^{-1}$** | **Factor** | **Above-Ground-P Uptake, kg ha$^{-1}$** |
| Treatment | CK | 13.2 ± 2.2 c | CK | 11.8 ± 1.8 c |
| | CFP | 19.8 ± 1.0 b | CFP | 18.2 ± 0.8 b |
| | LMP | 20.7 ± 0.1 b | LMP | 19.4 ± 0.6 b |
| | LMN | 19.3 ± 1.1 b | LMN | 18.5 ± 0.4 b |
| | SMP | 20.1 ± 1.1 b | SMP | 19.8 ± 0.9 b |
| | SMN | 25.5 ± 0.8 a | SMN | 24.1 ± 2.7 a |
| Year | 2004 | 20.6 ± 1.4 a | 2005 | 18.1 ± 1.5 ab |
| | 2006 | 19.6 ± 1.1 a | 2007 | 16.6 ± 0.8 b |
| | 2008 | 16.4 ± 1.1b | 2009 | 17.9 ± 0.9 ab |
| | 2010 | 20.1 ± 1.0 a | 2011 | 19.7 ± 0.8 ab |
| | 2012 | 22.1 ± 1.0 a | 2013 | 20.8 ± 1.5 a |

Means followed by the same letter within a column are not significantly different at $p \leq 0.05$ level.

Grain P removal was the product of grain yield and GPC. There was a study that indicated the significantly positive effect of P fertilization on GPC under long-term corn-soybean rotation [63]. In this study, the effect of P fertilization on GPC was also confirmed by the positive correlations between the GPC in corn and the P application rate ($p \leq 0.001$, *n* = 90) and between the GPC in soybean and the residual P for soybean after corn removal the previous season ($p \leq 0.001$, *n* = 90) for the corn-soybean rotation. P concentration in soybean grain was higher than corn. The plant growth/yield is affected by nutrient concentration in the tissue [57]. In this experiment, there is an obvious tendency to decrease for corn yield when GPC > 2.0 g kg$^{-1}$ (Figure 5). The GPC for SMN was the highest in

three of five corn years among treatments; the higher GPC corresponded to the relatively lower level of corn yield for SMN. The averaged GPC of LMN was higher than that of LMP in 2004, 2006, 2010 and 2012. This might have accounted for the less average yield with LMN than those with LMP. A decreasing trend for soybean yield was also presented when GPC was higher than 6.0 g kg$^{-1}$ (Figure 5). Römheld [64] considered 5.1~8.0 g kg$^{-1}$ P in soybean plant tissue as a high critical toxicity concentration for there would be a risk of growth reduction by inducing a deficiency of other nutrients. In addition, concentration effects may have contributed to higher GPC when crop yields were low.

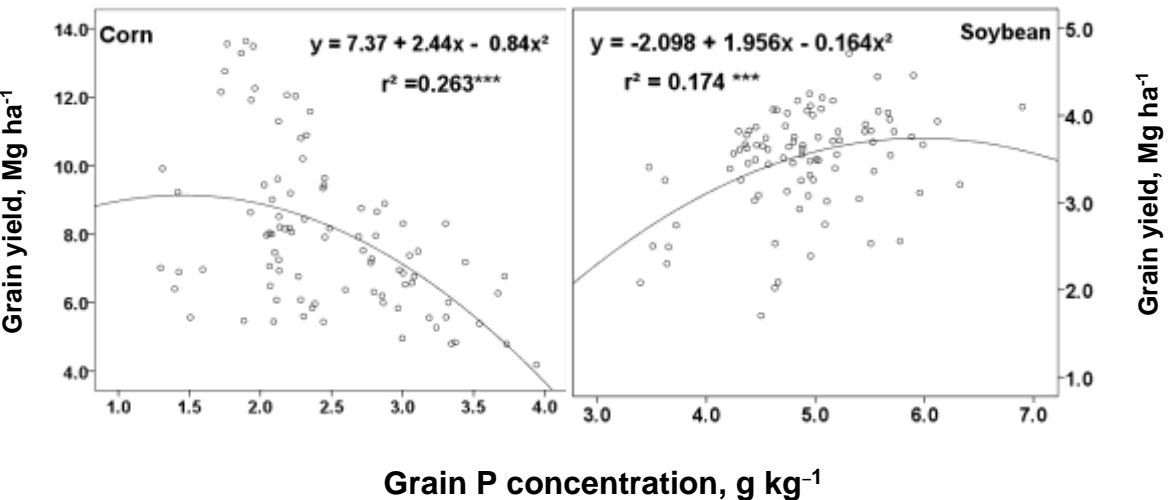

**Figure 5.** Relationship between grain P concentration (GPC) and grain yield for corn and soybean across five corn-soybean rotations (2004–2013) in a clay loam soil, Woodslee, ON, Canada. ***, significant at $p \leq 0.001$.

Grain P removal combined the effects of treatments on grain yield and P concentrations in grain. The GPC for SMN was the highest among treatments in 2004, 2008 and 2010 for corn and in crop years except 2013 for soybean (Data not shown). Large portion of P taken up by corn plant is removed in the harvested grain [65]. GPR accounts 70~88% and 77~86% of AGPU for corn and soybean, respectively. This indicates that significantly higher AGPU in SMN for corn was related to the contribution of high GPC, although the grain yield of SMN was low. The same reason can be used to explain the remarkably higher GPR and AGPU by the GPC in SMN for soybean.

During crop years, for corn, GPR in 2012 was significantly greater than other years and both GPR and AGPU in 2008 were significantly less than the other four years. For soybean, grain P removal in 2011 was significantly higher than that in the other cropping years except in 2013, as AGPU in 2013 was higher than other years and significantly higher than that in 2007 (Table 4). The higher yield of corn in 2012 and soybean in 2011 can partly explain grain P removal in 2012 and 2011 for corn and soybean, respectively.

### 3.3. Postharvest STP Content in the Soil Profile

Over the ten years of corn-soybean rotation, STP concentrated in the top 30 cm depth of the soil profile, while very low levels were detected below the top 30 cm soil layer for all treatments (Figure 6). In 0–7.5, 7.5~15 and 15~30 cm soil layers, STP for SMN were significantly higher than other treatments ($p \leq 0.05$) (data not shown). Compared with CK, the STP for SMN increased by 47.26, 48.17, and 29.63 mg kg$^{-1}$ in 0–7.5, 7.5~15 and 15~30 cm soil layers, respectively. The increment of STP in the three soil layers for LMN was second highest among treatments, were 9.58, 12.59, and 3.80 mg kg$^{-1}$, respectively. After ten years of biennial application, more STP remained in 0- to 30-cm soil layer in N-based swine manure treatments, especially in the SMN as compared to the chemical and P-based swine manure amendments. This result agreed with several other studies,

in which, when manure was applied based on crop-N requirements or at greater rates, P accumulation in soil occurred [66–68].

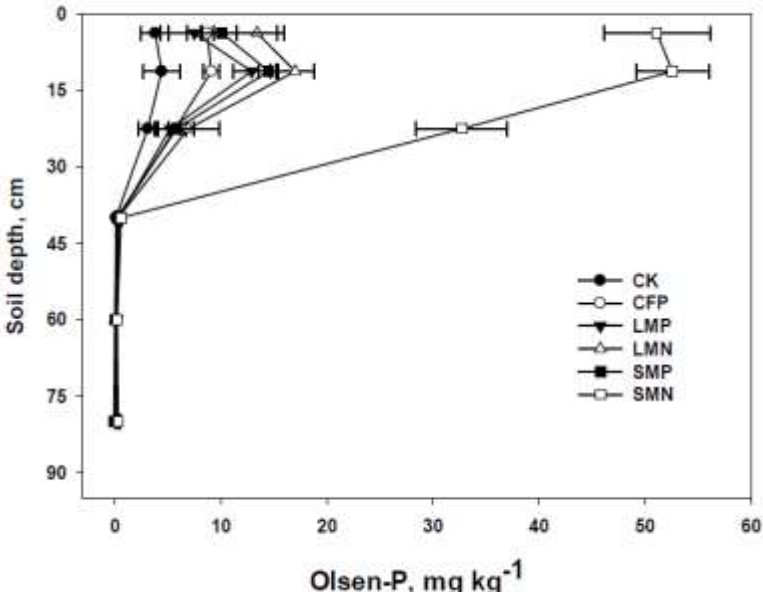

**Figure 6.** Postharvest soil test P (Olsen-P) content in 0–90 cm soil profile in response to various sources of P addition (chemical fertilizer vs. liquid and solid swine manures) and application approaches (P-based vs. N-based) after 10-year corn-soybean production in a clay loam soil, Woodslee, ON, Canada. Horizontal bars are standard errors (*n* = 3).

At the similar total P loading rates, the STP for LMP was only higher than that of the CK and lower than that of the other four treatments including the CFP in 0–7.5 and 15~30 cm soil layers. The STP for SMP was the second lowest among manure treatments. Relative to the STP prior to the initiation of the experiment (soil test P: 7.1 mg kg$^{-1}$), the increment of STP with LMP was the lowest among all manure treatments, which implies less potential environmental impact from agricultural land soil after long-term application versus other manure treatments. This is important for producers and nutrient management planners to make prudent management choices and to be mindful of the environment.

A forty-year study in western Nebraska found greater available P with manure application than with chemical fertilizer in a soil depth of 1.8 m [69]. The enhanced solubility of manure P possibly caused by P moved in organic forms or chemical reactions of P occurred with compounds in manure [69]. In the current experiment, in the deeper soil depth of 15–30-cm, except the SMN (32.7 mg kg$^{-1}$), STP for the CFP, LMP, LMN and SMP were close to each other with values of 5.7, 5.2, 6.9, and 5.8 mg kg$^{-1}$, respectively. The soil test P in 7.5–15 cm soil depth was the greatest among the six soil sections of soil sample column for both manure and chemical treatments (Figure 6). Crop roots resulted in effective tubular voids which contributed to the hydrodynamic behavior of topsoil [70]. A four-yr field experiment on the effects of P-based application of cattle manure under a corn–soybean rotation in the Lake Erie basin confirmed that the P in solid cattle manure was less prone to P loss than that in liquid cattle manure and inorganic fertilization after land application [71]. From this, the increased STP in 7.5–15 and 15–30 cm in SMN might not necessarily increase P loss from soil. Further research is needed on P leaching loss in association with N-based solid swine manure application.

After ten years of biennial application, the sum of mass of STP in top 30 cm soil layer (SMS 0–30 cm) for N-based manure treatments was greater than that for the P-based ones. The SMS (0–30 cm) for SMN (173.7 kg ha$^{-1}$) was significantly higher than other P treatments (Figure 7). The SMS (0–30 cm) for LMP (31.6 kg ha$^{-1}$) was close to that for CFP (30.1 kg ha$^{-1}$). Phosphorus based treatments for both liquid and solid swine

manure provided nutrients for crop growth while maintaining soil P level less than N-based treatments after ten years of application (Figures 1, 6 and 7). An eight-year-long research in Nebraska showed that P-based manure or compost application resulted in similar grain yields to N-based treatments but significantly less soil available P level [14,44]. More STP retained in the plot soil surface layers of SMN can cause a greater concern for P buildup than the other treatments. Phosphorus fertilization could increase grain yield in soils with low STP but not in soils with high STP [62], as shown in SMN with the high application rate of nutrient and high soil test P but less grain yield (Figures 1, 6 and 7).

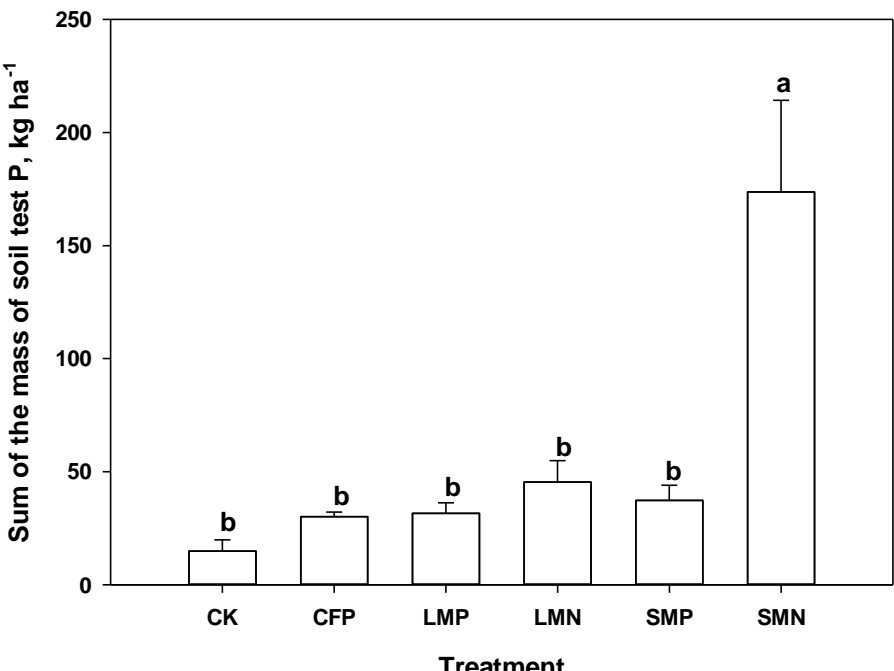

**Figure 7.** Effects of various sources of P addition (chemical fertilizer vs. liquid and solid swine manures) and application approaches (P-based vs. N-based) on the sum of mass of STP (Olsen-P) in top 30 cm plot soil (SMS 0–30 cm) after 10-year corn-soybean production in a clay loam soil, Woodslee, ON, Canada. Horizontal bars are standard errors ($n = 3$), the letters over the bars indicate differences at the significant level of $p \leq 0.05$.

### 3.4. Manure Phosphorus Source Availability Coefficients

For P-based manure treatments, the PSAC values averaged at 1.06 (range: 1.04–1.08) for liquid manure and at 1.08 (range: 0.98–1.13) for solid manure in both 0- to 15 and 0- to 30-cm soil depths, respectively (Table 5). The values of PSAC were close to 1.00 for LMP and SMP, suggesting a similar effect of P from liquid and solid swine manure under a P-based mode to those of the chemical fertilizer on P availability indicated in the sum of crop uptake and legacy STP in soils after 10 years of biennial application.

**Table 5.** Manure P source availability coefficient (PSAC) in the 0–15 and 0–30 cm soil depths after ten years (2004 to 2013) liquid and solid swine manure application based on crop P needs under corn-soybean rotation in a clay loam soil, Ontario, Canada.

|  | Liquid Manure | | Solid Manure | |
| --- | --- | --- | --- | --- |
| Cropping year | 0–15 cm | 0–30 cm | 0–15 cm | 0–30 cm |
| 2013 | 1.06 (0.01) [†] | 1.06 (0.01) | 1.07 (0.03) | 1.08 (0.05) |

[†] Numbers in brackets are SDs.

There are other methods on P efficiency for the long-term-consuming procedure, like the balance method, which was recommended by taking all applied P pools to account

for P removal by crop [41,43], but there may be cumulative P legacy effects from long-term application. During the long-term successive P addition through manure treatments, continuous crop removal of soil residual P resulted in decreases in labile inorganic P (Olsen-P) and increases in moderately stable P [72]. Residual P converts rapidly and primarily to P forms that are Olsen solution extractable and slowly to moderately labile P forms in soil that is NaOH (0.1 M) extractable [72,73]. For long term manure application, the magnitude of PSAC values implied a dual effects of manure treatments on supplying available P to crops and to soil P losses in various pathways, including surface runoff and tile drainage [74,75].

In recent years, phosphorus source coefficient (PSC) is introduced as a weighting factor for P site index (PSI) to account for the differential potential of P releasing to soil runoff after the application of various organic P sources in USA [76–79] and Canada [8,80]. The PSC was determined by soil testing in combination with either laboratory incubation [8,76–80] or laboratory incubation along with field validation studies with no crops grown [8,80]. A higher PSC value for liquid swine manure was reported relative to other major livestock manures [8,80]. Few studies on PSC have involved solid swine manure. For cropland soil, however, plant cultivation and nutrient management can increase P sorption near the soil surface, modify P distribution in soil solution, and lower P exports [81,82], which was consistent with the result under P-basis mode for liquid and solid manure application in the long-term corn-soybean rotation in current study. Phosphorus availability from soil-applied composts and manure is not only important for environmental but also for agronomic reasons. For example, the cumulative phosphorus uptake (CPU) of a crop was used to directly to measure P availability from soil-applied manures and composts in pot and field experiments [81,83]. Therefore, the PSAC seems to have more practicability and comprehensiveness in estimating the effect of long-term manure application on P availability by integrating phosphorus source, soil properties and bioavailable P for crop compared with PSC and CPU, and might be a universal index for both agronomic calibration and environmental risk indication to evaluate the long-term sustainability of fertilizer practices.

## 4. Conclusions

For the 10-year biennial application under corn-soybean rotation, liquid swine manure treatments often showed greater corn yields than solid manure treatments, and the corn yield the LMP was higher than SMP and SMN. The N-based strategy for solid swine manure resulted in a substantially higher amount of P applied, which did not bring about high yield. On the contrary, excessive P residue in the soil might pose increased risk contaminating the water resources.

After ten years of biennial application, the increment of the STP (relative to that of the CK) for N-based swine manure application treatments were higher than chemical fertilizer and P-basis manure treatments in 0–30 cm soil depth, especially for SMN which was significantly higher than all the other treatments. The increment of STP for LMP, on the contrary, was even less than that for CFP in 0–7.5 and 7.5–15 cm soil layers. Phosphorus-based swine manure application showed a greater benefits in both agronomic and environmental implications while soil P buildup from N-based manure application may become an environmental concern. PSAC appears to be a useful practical approach to evaluate both the agronomic and environmental effects of manure addition by integrating the P availability to crops in the year of application and the legacy the following years.

**Author Contributions:** Conceptualization, T.Z.; methodology, T.Z. and Y.W.; software, L.Z., X.H. and X.T.; validation, Y.W. and Y.Z.; formal analysis, Y.Z., T.W. and Y.W.; investigation, Y.W., T.Z. and Y.Z.; resources, T.Z., X.H. and C.T.; data curation, Y.W.; writing—original draft preparation, Y.Z.; writing—review and editing, T.Z. and Y.W.; visualization, X.C. and X.T.; supervision, T.Z. and Z.W.; project administration, T.Z.; funding acquisition, T.Z., C.T., and Z.W. All authors have read and agreed to the published version of the manuscript.

**Funding:** This research received no external funding.

**Acknowledgments:** The senior author appreciates the financially supported by the Western Project of China Scholarship Council and the Joint Student Scholarship program between China Scholarship Council and Agriculture and Agri-Food Canada. Field and lab technical supports from B. Hohner, M. Reeb, G. Statsco and D. Lawrence from Harrow Research and Development Center, Agriculture and Agri-Food Canada are acknowledged.

**Conflicts of Interest:** The authors declare no conflict of interest.

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
