# Peer review of "Crop Production and Phosphorus Legacy with Long-Term Phosphorus- and Nitrogen-Based Swine Manure Applications under Corn-Soybean Rotation"

_agronomy, doi:10.3390/agronomy11081548_

Round 1

Reviewer 1 Report

I have checked the revision, and the authors have fixed my round 1's comments, but I have added other notes, some of them as comments while others have been highlighted with red color.
you can compare this file with your original one. I have fixed some by myself but still others should be fixed. (space between words, style, and others).
You have written the title of some figures and tables bold while others have not.

Author Response

All of them have been revised according to the requirements of reviewer 1

Reviewer 2 Report

The manuscript "Production and phosphorus legacy after long-term phosphorus- and nitrogen-based swine manure application under corn-soybean rotation" concerned an interesting experiment investigated the effect of P-based, in comparison with N-based, liquid and solid pig manure application, on the yield of corn and soybean, grain P removal and above-ground-P uptake over a long time period and under field condition.

The manuscript need few minor revisions:

Authors refering to corn and soybean in general, however they studied certain varieties. It might be taken into consideration that corn and soybean exhibit wide range of traits regarding studied properties, so it might be necessary to involve more varieties into testing for being able to draw general conclusions.

Additionally, it is not clear enough whether the same varieties were planted every year? A short description of the used varieties should be given.

In my opinion, it would be more accurate to present the amount of rainfall and temperaturÄ™ over a ten-day periods, because in this way we can more accurately characterize the course of humidity and thermal conditions in which plants actually developed. From the point of view of availability of water and nutrients for plants, not only the sum of precipitation is important, but also their distribution.

Please provide the 30 year mean for temperature and precipitation either in tables or in your text. Right now you are just comparing temperature and data among the 10 years but it would be more valuable to other researchers to know how each year related to the 30-year mean.

Line 126: [4747] ???

Line 157: „wkafter” ???

Line 244: The difference in grain yield between LMN and SMP was small and insignifficant, so there is no need to mention it. The same is for LMN and SMN.

Figure 2: letters indicating significant differences are difficult to read.

Figure 3. I suggest to increase the font size, because they are poorly visible.

Line 318: Table 11 ??

Table 6: Lack of letters indicating significant differences for corn grain P removal  

Line 397: Olsen P for SMN were significantly higher than other treatments – how do you know? Figure 3 does not show the results of statistical analysis.

Line 401: …. The increment were significantly higher than that of other four treatments…. – the same – Figure 3 does not contain information about significant differences.

Line 484: [7371] ???

Author Response

The manuscript "Production and phosphorus legacy after long
term phosphorus and
nitrogen based swine manure application under corn soybean rotation" concerned an
interesting experiment investigated the effect of P based, in comparison with N based, liquid
and solid pig manure application, on the yield of corn and soybean, grain P removal and
above ground P uptake over a long time period and under field condition.
The manuscript need few minor revisions:
Authors
refering to corn and soybean in general, however they studied certain varieties. It
might be taken into consideration that corn and soybean exhibit wide range of traits regarding
studied properties, so it might be necessary to involve more varieties into t esting for being able
to draw general conclusions.
Additionally, it is not clear enough whether the same varieties were planted every year? A short
description of the used varieties should be given.
Response: thank you for the careful checking! The varieties for corn and soybean have appeared in the abstract and been added in L147 and L158 in the revision.
In my opinion, it would be more accurate to present the amount of rainfall and temperaturÄ™
over a ten day periods, because in this way we can more accurately characterize the course of
humidity and thermal conditions in which plants actually developed. From the point of view of
availability of water and nutrients for plants, not only the sum of precipitation is important, but
also their distribution.
Please provide the 30 year mean for temperature and precipitation either in tables or in your
text. Right now you are just comparing temperature and data among the 10 years but it would
be more valuable to other researchers to know how each y ear related to the 30 year mean.
Response: That's a good suggestion! But the purpose of this paper is to explain the impact of fertilization strategies on crops. If the impact of weather conditions on crops is expounded every ten days, some analysis and discussion will be needed if the impact of weather conditions on crops is expounded over a ten-day period which may divert readers' attention from the impact of manure application strategies.
I tried to find the previous climate data before the experiment from the engineer and the project leader or form internet, but only the data for some intermittent terms could be found, the mean
of temperature and precipitation for the 30 years could not be gotten relying on these incomplete data.
Line 126: [4747] ???
Line 126: [4747] ???
Response: thank you for the careful checking! The correction has been made in L142 with a new reference index number [48] in the revision.
Line 157: „wkafter” ???
Line 157: „wkafter” ???
Response: thank you for the careful checking! The correction has been made in L161 in the revision.
Line 244: The difference in grain yield between LMN and SMP was small and insignifficant, so
Line 244: The difference in grain yield between LMN and SMP was small and insignifficant, so there is no need to mention it. The same is for LMN and SMN.there is no need to mention it. The same is for LMN and SMN.
Response: thank you for the kind reminder! The comparison between LMN and SMP, LMN and SMN has been deleted. The comparison between N- and P-based treatments has been moved to Line 242 in the revision.
Figure 2: letters indicating significant
Figure 2: letters indicating significant differences are difficult to read.differences are difficult to read.
Response: thank you for the kind reminder! The figure has been redone and the corn section in this figure has been changed into Figure 1 in the revision, as the corresponding author deemed that Figure1A was the repetition of Figure 2A in the original “Agronomy- 991171”, and Figure 2B was deleted because there was no significant interaction between year and treatment for soybean yield. Please see the writing ideas of the section “3.1. Crop yield” in revision by the item of L222-326 in “Section 5” of this letter.
Figure 3. I suggest to increase the font size, because they are poorly visible.
Figure 3. I suggest to increase the font size, because they are poorly visible.
Response: Figure 3. has been redone and has been changed into Figure 2 in the revision. Figure 3D, E, F was deleted because there was no significant interaction between year and treatment for soybean yield.
Line 318: Table 11 ??
Line 318: Table 11 ??
Response: thank you for the careful checking! The correction has been made in L291 in the revision.
Table 6: Lack of letters indicating significant differences for corn grain P removal
Table 6: Lack of letters indicating significant differences for corn grain P removal
Response: thank you for the careful checking! And the comparation with corn grain P removal has been showed in Figure 4 for the presentation in the graph enables the corn grain P removal in the way of treatment x year interaction. Table 6 in the original “Agronomy- 991171” has been changed into Table 4 in the revision by deleting the section of grain P removal and adding
with “± standard deviation” after the value of above-ground-P uptake in order to be more rigor and scientific. Please see the writing ideas of the section “3.2. Grain P removal and above-ground-P uptake” in the revision by the item of L327-392 in Section 5 of this letter.
Line 397: Olsen P for SMN were significantly higher than other treatments
Line 397: Olsen P for SMN were significantly higher than other treatments –– how do you know? how do you know? Figure 3 does not show the results of statistical analysis.Figure 3 does not show the results of statistical analysis.
Response: thank you for the kind reminding! A note of “Data not shown” in bracket in L401 has been added in the revision.
Line 401: …. The increment were significantly higher than that of other four treatments….
Line 401: …. The increment were significantly higher than that of other four treatments…. –– the the same same –– Figure 3 does not contain Figure 3 does not contain information about significant differences.information about significant differences.
Response: thank you for the careful checking and kind reminding! Now the sentence has been deleted, even though another figure was done to show this result, for the corresponding author thought that would be the repetition of Figure 6 (in the revision).
Line 484: [7371] ???
Line 484: [7371] ???
Response: The correction has been made in L477 in the revision.

Reviewer 3 Report

Please see the comments in the attached file.

Author Response

There are 8 pieces of comments before the following comments have been responded in the revision. The following responses will be showed in details.
1. L64-66 Elaborate more. Why does cattle manure lead to the buildup of P in the soil relative to swine manure? Also, provide more sources to back up that point.
Response: the reason was added in L80-82 and new references [27, 28] were added as well in the revision. The newly added parts have been highlighted in yellow.
2. L74-75 a brief about rotation was highlighted, can more be written about how rotation affects P accumulation in the soil – with sources to back it up?
Response: the reason on the declination of soil-test P on the rotated plots is given in L89-90 in the revision (in yellow).
3. L168-173 don’t you think this sentence is more appropriate under L121 where manure sampling and chemical compositions are highlighted? In addition, Table 2 cross-referenced on L173 recorded temperature. Are you referring to Table 3?
Response: thank you for the kind reminding! This is indeed my negligence. L168-173 was deleted for the same paragraph has appeared above.
4. L225 Don’t you think you performed a polynomial regression as opposed to a multiple linear regression?
Response: thank you for the kind reminder! The correction has been done in L215 in the revision.
5. L242-244 I think that sentence should be split into two parts, as it is not easy to follow in its present state e.g. yield for LMP was higher than that of SMP and SMN by 1.91 and 1.94 Mg ha -1 respectively. Similarly, yield for LMN exceeded that of LMP and LMB by …... The same sentence talks of annual yields, but isn’t the difference reported here is an average of the five or so corn experimental years? Is this yield difference significant or it is just due to random error?
Response: thank you for the comment! There seems to be something wrong with the narration of this sentence. The difference in grain yield between LMN and SMP was small and insignificant. The sentence has been replaced with new sentences in L242-246 to give comparisons on the annual corn grain yield between of N- and P-based treatment in the revision. Please see the writing ideas of the section “3.1. Crop yield” in the revision by the item of L222-326 in Section 5 of this letter.
6. L245 yield for LMN was no different from yield of SMN and SMP, why was that the case since yield for LMP was highly different from others?
Response: thank you for the question! The application rate of nutrient may be an influencing factor, as shown in previous studies (in L246-250 in the revision). The comparison between LMN and LMP was added in L242-243 in the revision (painted yellow).
7. L246-247 seems to be a repetition of a sentence found on L242-244. Why don’t you delete “The annual yields …...than that of SMP and SMN” and leave the remaining part with a bit of a modification?
Response: thank you for the kind comment! The sentence has been rewritten in L243-244 in the revision.
8. L249 begin a new paragraph for soybean reporting
Response: thank you for the kind reminder! A new paragraph (L311-320 in the revision) about soybean yield has been made by combing the sentences in L239-241, L249-255 and L319-321 in the original “Agronomy- 991171”, this is according to the comments of our team that the section “3.1. Crop yield” as showed in the item of L222-326 in Section 5 this letter.
9. 255-256 for reporting related to Figure 1, did you perform a single degree of freedom contrast? Multiple comparisons may not detect some differences between treatments that single DOF contrast is able to do. Without the data it is difficult to tell, but it is possible that with a single DOF contrast, corn yield for CFP may be statistically different from that of SMP and SMN.
Response: Thank you for this comment! We understand that it looks like there might be s significant difference. However, we conducted a single degree of freedom contrast. The results show that there was no significant difference between CFP against SMP and SMN. The p value was 0.1543 and 0.1877, respectively.
10. L269-270 why does it cause a decline in yield? Possibly due to N toxicity? Here is a source but from long time ago, if you are lucky to find a recent source, that will be good
https://acsess.onlinelibrary.wiley.com/doi/abs/10.2134/1990.nitrogenincropproduction.c6
Response: Thank you for the kind recommendation! A sentence in L250-252 was added in the revision according to the comments of our team. And in L369-371 in the revision, the discussion about the reason for the relatively lower corn yield for SMN was also added.
11. L271 figures are very small to make any meaningful conclusion from them. If you must show such, why not just present a table for yield in each rotation period with one additional column for rainfall?
Response: Thank you for this comment! The figure (Figure 3 in the revision) has been redrawn in L225 in the revision. For the significant interaction by treatment x year for corn grain yield needed to be presented in the way of graph. The figure has been redone in L225 by deleting the section of soybean for there is no significant interaction by treatment x year for soybean grain yield, as showed in Table 3 in the revision.
12. L280 I don’t think it is the effect of treatment since in each year, the same amount of nutrients were applied, leaving year as the sole independent variable. Residual P and K may not matter that much even if they accumulated in the soil since the yield of corn does not depend on the total amount present in the root system sorption zone but rather the concentration at or near the root surface
Response: thank you for the comment! The segment “effect of treatments on” has been deleted.
13. L282-283 I am not sure why you are beginning to look at the effect of treatment within each year again. I thought the overall effect of treatment previously reported was adequate and can be generalized regardless of the year. Why not just focus on the effect of year that you started above?
Response: thank you for the suggestion! I have a hesitation to delete these sentences, for the significant effect was showed for cropping year on corn yield in Table 3, so the comparisons among the yields of treatments was made here in L253-255 in the revision, and the following three paragraphs after these focused on the effect of year.
14. L304-305 change one of the months. It says “The average temperature in May ………was 1.7C higher than the average temperature in May.”
Response: thank you for the kind reminder! It's because of my carelessness. The corresponding years for the months were added in L271, 272 and 273 in the revision.
15. L319 start a new paragraph for soybean discussion
Response: thank you for the kind comment! All the discussions about soybean yield have been combined in the new paragraph (L311-320 in the revision) about soybean yield has been made by combing in L239-241, L249-255 and L319-321 in the original Agronomy- 991171, this is according to the comments of our team about section “3.1. Crop yield”, the idea was showed in the item of L222-326 in Section 5 this letter.
16. 322-323 rephrase that sentence. It is not clear when you say temperature in July and September was 1.5°C higher and yet at the same time indicating that it was 0.9°C lower. You
seem to mean temperature in July was 1.5°C higher than the average temperature (for the growing season?) while that of September was 0.9°C lower than the average temperature. The sentence ranging from L322 to L326 is simply too long that it would be appropriate to split it up.
Response: thank you for the comment! The rephrase has been done in L293-295 in the revision.
17. L337 Table 6 has under “Factor” heading a title “Treatment x year”. What is that one representing since the table does not show mean values for treatment by year interaction? The column under soybean that represents treatment and year is a repetition since that information is already presented in the first column of the table. What is the difference between ANOVA in Table 5 and Table 6 (tail of the table)? That seems to be a repetition. If that is correct, I would prefer replacing the information in Table 5 with the information at the tail of Table 6 since it is a bit more detailed e.g. giving DF, F value and actual p-value.
Response: thank you for the suggestion! The section about corn grain P removal has been transferred to the Figure 4 in the revision for the significant interaction by treatment x year for corn grain P removal was needed to be presented in the way of graph according to the comments of our team. Table 6 has been changed into Table 4 in the revision, and adding with “± standard deviation” for the value of above-ground-P uptake, actual p-value has been added in Table 3 (i.e., Table 5 in the original Agronomy- 991171) in the revision according to the comments of the team.
18. L343-345 Indicated that GPR was significant. Where are the letters to show that the CK was significantly lower than that of other treatment levels under corn GPR?
Response: thank you for the question! Now the information has been identified in the Figure 4 in the revision.
19. L397 significant at what probability level?
Response: thank you for the kind reminder! The probability level of significance has been added in L401 in the revision.
20. L403, “except SMN”? We already know that in your previous statement. Unless reference is made to another treatment within the 0-30 cm depth, I suggest deleting that part of the sentence.
Response: the sentence L401-403 including “except SMN” has been deleted according to the comments of the team for the result of the comparation among treatments was not presented in Figure 6.
21. L412 what does “Relative to the original soil previous the experiment” mean? Rephrase
Response: thank you for the kind comment! The rephrasing has been made in L411 in the revision.
22. L429 “sum of mass” of? The sentence itself appears to be incomplete. What happened to the sum of mass in the top 30 cm?
Response: thank you for the careful checking! This sentence has been replenished in L438-439 in the revision.

Reviewer 4 Report

Overall comments:

Interesting study in terms of long-term data and focus on sustainability. This paper needs work in flow and English comprehension, specifically in the Introduction section. Results could use some more justification, specifically in section “3.3 Postharvest STP Content in the Soil Profile”. Figure 2, Figure 3, and Figure 4, are unclear, small, or low quality and need improvement. This paper would benefit from a more well-rounded conclusion and implications for use of the findings. 

Specific comments:

Line 23: Unclear what LMN, SMN, LMP, and SMP mean. Maybe add abbreviations next to the words liquid and solid respectively.  

Line 29: Correct “closed” to “close”

Line 32: Correct agronomy to agronomic

Line 52: Head of pigs?

Line 53: Remove “effecting”

Line 64-66: Unclear sentence. Needs rewording to correct English.

Line 67-70: May be better suited to be added to discussion or summarize point into a more concise sentence.

Line 73: Change “more available” to “more plant available”

Line 81-82: Reword to make a clear point

Line 89-93: Reword to non-question format or remove

Line 95: Change “taken away by crops” to “utilized by crops” or “taken up by crops”

Line 117-120: Temperature and precipitation tables may be better represented as graphs

Line 154–155: Did disking occur for all treatments simultaneously?

Line 168-170: This sentence is confusing, reword.

Line 170: Do not begin sentence with number “2”, spell it out.

Line 174: Change beginning of sentence to “Each year at harvest time,”

Line 194: Change “Meanwhile also,” to “Simultaneously”

Line 197-199: Was all bulk density data non-significant?

Line 244-246: Edit to read something like:  

“It is possible that more time was needed for microbial decomposition and cycling of nutrients in solid manure than in liquid manure. As such, this phenomenon may confine the release of nutrients in solid manure treatments”

Line 252: change to “treatments were slightly influenced by the residual soil P…”

Line 264-269: Run on sentence, break up for easier read.

Line 271: Figure is blurry, significance indicators and legends are hard to read. At first glance, meaning of top bars is confusing. An explanation should be added in the figure caption to explain seasonal rainfall indicators.

Line 285: Figure is very small and hard to read.

Line 290: Change to “A study conducted in southeast Nebraska”

Line 290-294: Another run-on sentence. Break up into multiple sentences for better flow.

Line 294-298: Same as previous comment.

Line 305-306: Confusing, do you mean 3-day average temperature?

Line 460: Replace “closed” with “close”.

Line 498: Same as previous comment.

Line 505: What are the implications of this study? How can the conclusions made based on results be utilized by farmers in the Lake Erie Basin or in similar farming systems?

Author Response

Overall comments:
Interesting study in terms of long
term data and focus on sustainability. This paper needs work
in flow and English comprehension, specifically in the Introduction section. Results could use
some more justi fication, specifically in section “3.3 Postharvest STP Content in the Soil Profile”.
Figure 2, Figure 3, and Figure 4, are unclear, small, or low quality and need improvement. This
paper would benefit from a more well rounded conclusion and implications fo r use of the
findings.
Specific comments:
Line 23: Unclear what LMN, SMN, LMP, and SMP mean. Maybe add abbreviations next to the
words liquid and solid respectively.
Response: Thank you for the suggestion! The explanatory note for the abbreviations of LMN, SMN, LMP, and SMP have been added in L24-25 in the revision.
Line 29: Correct “closed” to “close”
Response: Thank you for the comment! The correction has been made with “largely identical” replace “closed” the in L39 in the revision.
Line 32: Correct agronomy to agronomic
Response: Thank you for the comment! The correction has been made in L44 in the revision., even though the rest of the sentence has been rephased.
Line 52: Head of pigs?
Response: “Head” has been deleted in the revision.
Line 53: Remove “effecting”
Response: Thank you for the comment! The “effecting” correction has been removed in the revision.
Line 64
66: Unclear sentence. Needs rewording to correct English.
Response: Thank you for the comment! The rewording has been done in L75-78 in the revision.
Line 67
70: May be better suited to be added to discussion or summarize point into a more
concise sentence.
Response: Thank you for the comment! The rewording has been done in L82-84 in the revision.
Line 73: Change “more available” to “more plant available”
Response: Thank you for the comment! “plant” has been added in L86 in the revision.
Line 81-82: Reword to make a clear point
Response: Thank you for the comment! The rewording has been done in L95-97 in the revision.
Line 89-93: Reword to non-question format or remove
Response: The question format (Line 103-107 in revision) was suggested by the corresponding
author, for it is a good form to put forward the problems to be studied in this paper.
Line 95: Change “taken away by crops” to “utilized by crops” or “taken up by crops”
Response: Thank you for the comment! The rewording has been done in L109 in the revision.
Line 117-120: Temperature and precipitation tables may be better represented as graphs
Figure 1 Growing seasonal monthly precipitation on the experimental site from 2004 to 2013.
Month
March April May June July Aug. Sept. Oct. Nov.
Monthly precipitation, mm
0
50
100
150
200
250
2004
2005
2006
2007
2008
2009
2010
2011
2012
2013
cropping year
2002 2004 2006 2008 2010 2012 2014
Temperature, ℃
0
5
10
15
20
25
30
March
April
May
June
July
Aug.
Sept.
Oct.
Nov.
Figure 2 Growing seasonal monthly temperature on the experimental site, Woodslee, ON, Canada, from 2004 to 2013.
Response: Thank you for the comment! The two graphs on the monthly precipitation and temperature were not put forward to the revision for them looks a little messy as showed in Figure 1 and 2 in this cover letter.
Line 154
Line 154––155: Did disking occur for all treatments simultaneously?155: Did disking occur for all treatments simultaneously?
Response: Thank you for the question! The words “by disking to a soil depth of 15 cm” was deleted according to the corresponding author comment.
Line 168
Line 168--170: This sentence is confusing, reword.170: This sentence is confusing, reword.
Response: Thank you for the comment! The sentence has been reworded in L137-138, for this sentence was the repeated in two places in the original version, in L122-126 and L168-173, so the sentence in L168-170 has been deleted in the revision.
Line 170: Do not begin sentence with
Line 170: Do not begin sentence with number “2”, spell it out.number “2”, spell it out.
Response: Thank you for the comment! The rewording has been done in L138 in the revision.
Line 174: Change beginning of sentence to “Each year at harvest time,”
Line 174: Change beginning of sentence to “Each year at harvest time,”
Response: Thank you for the comment! The rewording has been done in L173 in the revision.
Line 194: Change “Meanwhile also,” to “Simultaneously”
Line 194: Change “Meanwhile also,” to “Simultaneously”
Response: Thank you for the comment! The rewording has been done in L189 in the revision.
Line 197
Line 197--199: Was all bulk density data non199: Was all bulk density data non--significant?significant?
Response: Thank you for the comment! Line 195-199 in original Agronomy- 991171 was deleted according to the comments of the team for they consider these sentences could dilute the focuses of this paper.
Line 244
Line 244--246: Edit to read something like: 246: Edit to read something like:
“It is possible that more time
“It is possible that more time was needed for microbial decomposition and cycling of nutrients was needed for microbial decomposition and cycling of nutrients in solid manure than in liquid manure. As such, this phenomenon may confine the release of in solid manure than in liquid manure. As such, this phenomenon may confine the release of nutrients in solid manure treatments”nutrients in solid manure treatments”
Response: Thank you for your kind comment! According to the comments of the team, the reason for less yield for SMN is as showed in L235-237 in the revision.
Line 252: change to “treatments were slightly influenced by the residual soil P…”
Line 252: change to “treatments were slightly influenced by the residual soil P…”
Response: Thank you for the kind comment! The addition has been done in L315 in the revision for the discussion about soybean yield has been placed together. Please refer to the item of L222-326 in Section 5 in this letter.
Line 264
Line 264--269: Run on sentence, break up for easier read.269: Run on sentence, break up for easier read.
Response: Thank you for the kind comment! The rewording has been done in L246-250 in the revision. The discussion about corn and soybean yield were separated in two parts in the revision according to the comments of our team.
Line 271: Figure is blur
Line 271: Figure is blurry, significance indicators and ry, significance indicators and legends are hard to read. At first glance, legends are hard to read. At first glance, meaning of top bars is confusing.meaning of top bars is confusing. An explanation should be added in the figure caption to An explanation should be added in the figure caption to explain seasonal rainfall indicators.explain seasonal rainfall indicators.
Response: Thank you for the kind comment! The figure (Figure 3 in the revision) has been redrawn in L225 in the revision. For the significant interaction by treatment x year for corn grain yield needed to be presented in the way of graph. The figure has been redone in L225 by deleting the section of soybean for there is no significant interaction by treatment x year for soybean grain yield, as showed in Table 3 in the revision. The explanation has been added in the figure to indicate seasonal rainfall.
Line
Line 285: Figure is very small and hard to read.285: Figure is very small and hard to read.
Response: Thank you for the comment! Figure 3. has been redone and has been changed into Figure 2 in the revision, and Figure 3D, E, F was deleted because there was no significant interaction between year and treatment for soybean yield. The figure has been redrawn in L278 in the revision.
Line 290: Change to “A study conducted in southeast Nebraska”
Line 290: Change to “A study conducted in southeast Nebraska”
Response: Thank you for the kind comment! The rewarding has been done in L257 in the revision.
Line 290
Line 290--294: Another run294: Another run--on sentence. Break up into multiple sentences for better flow.on sentence. Break up into multiple sentences for better flow. Response: Thank you for the kind comment! The rewarding has been done in L257-261 in the revision.
Line 294
Line 294--298: Same as previous comment.298: Same as previous comment.
Response: Thank you for the kind comment! The rewarding has been done in L261-266 in the revision.
Line 305
Line 305--306: Confusing, do you mean 3306: Confusing, do you mean 3--day average temperature?day average temperature?
Response: thank you for the careful checking! The clearer sentence has been written to replace the original one in L274-275 in the revision.
Line 460: Replace “closed” with “close”.
Line 460: Replace “closed” with “close”.
Response: thank you for the careful checking! The sentence with this word has been deleted in the revision.
Line 498: Same as previous comment.
Line 498: Same as previous comment.
Response: thank you for the careful checking! The correction has been made in L518 in the revision.
Line 505: What are the implications of this study? How can the conclusions made based on
Line 505: What are the implications of this study? How can the conclusions made based on results be utilized by farmers in the Lake Erie Basin or in similarresults be utilized by farmers in the Lake Erie Basin or in similar farming systems?farming systems?
Response: it is a good question! The implications of this study located in L42-44, the conclusions for the utilization by farmers in the Lake Erie Basin or in similar farming systems were in L490-491 and L497-499
Section 5 The explanation about the parts
Section 5 The explanation about the parts revised revised according the according the team and team and corresponding author corresponding author commentscomments::
L28-34 in the revision (painted in yellow) was added according the corresponding author comments for providing more details to present the effects of P-based vs. N-based.
L41-44 in the revision (painted in yellow), the revised parts according the corresponding author comments.
L111-113, in the revision (painted in yellow), the revised parts according the corresponding author comments.
L142-144, in the revision (painted in yellow), the revised parts according the corresponding author comments.
L142, Table 3 in the original “Agronomy- 991171” has been deleted and replaced by the sentence in L142-144 in the revision, for according to the corresponding author comments, the application rates of manures for each the treatment were determined also by the contents of moisture except the available N and or total P.
L156, Table 4 in the original “Agronomy- 991171” has been deleted according to the corresponding author comments and the team, for the content, i.e., the addition way for three P sources, has been narrated in the first paragraph in “2.3. Experimental Design”. And L150-152 in the original “Agronomy- 991171” has also been deleted from the corresponding author
comments for all things in this paragraph have been mentioned above.
L193-194 in the revision (painted in yellow), the revised parts according the corresponding author comments.
L204-208 in the original “Agronomy- 991171” has been deleted according to the corresponding author comments for this part is little to do with the content of the manuscript.
L222-326 in the revision, the section “3.1. Crop yield” has been revised as the comments from the team and the corresponding author comments as the outlines as the followings:
1. Corn yield - interactions between treatment and year, as your stats show the significance. Discussion should be specifically focused on your data.
2. Corn yield - comparisons between P-based vs. N-based manure application;
3. Soybean yield – focusing on the individual effects of treatment and the year, as there were not interactions between the two factors; and
4. Soybean yield - comparisons between P-based vs. N-based manure application.
L225 “Figure 1” in the original “Agronomy- 991171” has been changed into the current one in revision for the corresponding author deemed that Figure1A was the repetition of Figure 2A in the original “Agronomy- 991171”, and Figure 2B was deleted because there was no significant interaction between year and treatment for soybean yield.
L229-237 in the revision (painted in yellow), the revised parts according the corresponding author comments.
L239-241 in the revision (painted in yellow), the revised parts according the corresponding author comments.
L245-246 in the revision (painted in yellow), the revised parts according the corresponding author comments.
L281, Figure 2 has been redone in the revision. Figure 3D, E, F in the original “Agronomy- 991171” were deleted because there was no significant interaction between year and treatment for soybean yield.
L293-295 in the revision (painted in yellow), the revised parts according the corresponding author comments.
L303, Figure 3 has been redrawn by deleting the section of corn yield.
L311-320, this new paragraph about soybean yield was made by combing in L239-241, L249-255 and L319-321 in the original “Agronomy- 991171”, this is according to the comments of our team that the section “3.1. Crop yield” should include 4 parts, as showed in the
item of L222-326 in this section.
L327-392 The section “3.2. Grain P removal and above-ground-P uptake” has been revised as the comments as following:
Present the results and make discussion following the order as shown as below,
1. GPR for corn - interactions between treatment and year, as your stats show the significance;
2. Corn GPR - comparisons between P-based vs. N-based manure application (i.e. SMP vs. SMN; LMP vs. LMN);
3. AGP for corn and soybean – effects of treatment and year, respectively; comparisons between P-based vs. N-based manure application (i.e. SMP vs. SMN; LMP vs. LMN);
L330, Figure 4 was added according the corresponding author comments to show the significant interaction by treatment x year on corn grain P removal in the way graphed for there was an interaction between treatment and year for corn grain P removal, as showed in Table 3.
L360-361, Table 4 in the revision was from Table 6 in the original “Agronomy- 991171” by deleting the section of grain P removal and adding with “± standard deviation” for the value of above-ground-P uptake in order to be more rigor and scientific.
Figure 7 in the original “Agronomy- 991171” was deleted for the relationships aren’t strong, although they are statistically significant. There is lack of accountability for the legacy effect of P applied.
L363-367, in the revision (painted in yellow), the revised parts according the corresponding author comments.
L369-371, in the revision (painted in yellow), the revised parts according the corresponding author comments.
L387-388, in the revision (painted in yellow), the revised parts according the corresponding author comments.
L417-420, in the revision (painted in yellow), the revision was done by reversing the sentence in L417-418 in original “Agronomy- 991171” with the one next after it.
L423-430, in the revision (painted in yellow), the revised parts according our teamr comments.
L452-455, in the revision (painted in yellow), the revised parts according the team comments.
L456-459, Table 5 was added to the details of the PSAC values for liquid manure and solid manure in 0- to 15 and 0- to 30-cm soil depths, respectively).
L479-485, in the revision (painted in yellow), the revised parts according the team comments.
L494-502, in the revision (painted in yellow), the revised parts according our team comments.
L587-591, the reference was added as the response to Reviewer 3 comments.
L657-665, L700-706 and L710-712, the references were added as the corresponding author comments.

This manuscript is a resubmission of an earlier submission. The following is a list of the peer review reports and author responses from that submission.

Round 1

Reviewer 1 Report

General comments:

The purpose of the study is to evaluate the long-term effects of manure applied at different rates – to supply the full nitrogen needs of the corn soybean rotation (which overapplies phosphorus) or to supply the full phosphorus needs of the corn-soybean rotation (and supplement the remaining N needs with fertilizer). It is not a new concept but the changing climate in many regions has prompted these types of studies again to re-evaluate the impacts on crop yields and soil nutrient cycling. The added benefit of this study is its long-term nature.

The experiment seems to be well designed except that initial soil samples to 90 cm depth were not collected to compare with the end of the study samples and the authors chose to use the Olsen soil test for phosphorus, which is very unusual for an acidic soil. The test itself was designed specifically for carbonate soils and this study was not conducted on a carbonate soil. While relative comparisons can be made across treatments, the results are not widely applicable and reliable comparisons to most other studies cannot be made. Ideally the soil samples would be analyzed with a more appropriate test and the paper modified with the new results. Otherwise, editing the results section to discuss some of these implications is a must.

There are also a few other significant issues that should be addressed. More details are below.

Specific comments (line numbers refer to those seen in the PDF version downloaded from the MDPI reviewer system):

Line 41: I’m not sure what a P fossil is in this context, were the authors referring to rock phosphate?

Lines 57-58: This is unclear. Is the N:P ratio 1.46:2.88 or are the authors suggesting the range is 1.46:1 to 2.88:1?

Line 111: Why was Olsen P used on an acidic soil? This test is usually reserved for neutral to acidic soils. Is the classification of "low" applicable here if the test is not designed for this type of soil? If this test was calibrated for crop yields within this pH range, then please cite a reference.

Table 1: Why is only precipitation included in this table? Please also include average temperatures.

Lines 120-125: There is a lot of missing key information here. What were the availability factors used for N, P, and K in the different manure types? Or were they the same? Please also include the following information (perhaps in a table): application rates of each manure type each year and the assumed available N, P, and K from manure that was added each year plus fertilizer N, P, and K. As written, it is unclear whether nutrients were balanced in each treatment.

Lines 126-127: Fertilizers and manure were only added in the corn year phase. Was this sufficient P and K for the soybean phase of the rotation? Please indicate if this is the case. I would also suggest adding the recommended N, P, and K levels for your region to give readers and idea of whether nutrient needs were actually being met.

Lines 135-136: Because this is an international journal, it may be useful to share the pesticide/herbicide regime in case others are not familiar with corn/soybean production in this region.

Table 2: Thank you for including total N, P, and K in the manure for each year. Please add the suggested information as described in my comments above (estimated available N, P, and K added each year to each treatment with manure and fertilizer listed separately).

Line 143: How was the liquid swine manure collected? Where was it stored? In a pit? In an earthen basin? Was it agitated prior to being collected?

Line 144: What kind of operation was liquid swine manure collected from? Finisher barn? Nursery barn? Wean to finish? The nutrient content of manure from different kinds of operations varies greatly.

Lines 145-146: What is meant by manure was collected from a “pile”? Was this a stockpile on the farm site? How long had the manure been stockpiled?

Lines 146-147: When it says "samples were collected 2-3 days prior before application" does that mean the actual manure that was to be used in the study was collected then, or was manure collected earlier and only subsamples were collected 2-3 days before application?

Line 148: It is unclear why there were different number of rows for the different crops. Are narrow-row soybeans typically practiced in this region? Please specify for readers who are not familiar with cropping systems in this area.

Lines 151-152: This "grain analyzer" measured yield or grain moisture? The way this sentence is written makes it unclear. I assume the grain analyzer measured moisture content and yields were adjusted to 15.5 and 13% moisture?

Lines 160-161: Soil samples to 90 cm of depth were only collected at the end of the experiment? Why were they not collected at the beginning, too, as a comparison over time?

Lines 162-162: It is unclear where these extra samples were coming from. Each plot? Were all these samples combined for a plot for each depth then subsampled for nutrient analysis or was each individual sample analyzed? Please clarify the sampling procedure.

Line 164: I would not expect the Olsen P test to work well with soils with a pH of 6.2 since the soil would likely neutralize some of the extractant, thus underestimating P levels. It was designed for calcareous soils which have alkaline pH. While relative comparisons can probably be made across treatments within this study, comparisons to other studies cannot be made.

Line 171: Is the SMS the mass of the P in the top 90 cm of soil? Or from a different soil layer? Please clarify.

Line 172: Bulk density was measured? By what method? For each plot? This was not described in the methodology and should be added. Were their differences across treatments due to the different manure sources? This information should probably be included.

Equation in lines 175-176: Is CF in the denominator supposed to be CFP? I do not think CF was a treatment.

Line 179: This was not considered a mixed model then? Replications were considered as fixed effects? Please clarify or consider using mixed model techniques. Also add whether data was evaluated for normality and if/how non-normality was dealt with.

Lines 200-201: Why is that? There doesn't seem to be any discussion of this. I suspect it may relate to differences in nutrient release between the two manure types. What N and P availability factors were used for these two different manure types? Were they they same or were they different? This was not described in the methods.

Lines 202-204: It is unclear why the LMP yields were higher than the LMN yields if the available N rates were in fact, equal. Please provide enough information in the methodology to determine this. In the discussion, please provide some reasoning as to why this might have occurred.

Lines 219-223: The article cited here (Eghball et al.) did not have the results or the interpretation that the authors described. For example, they found that P-based and N-based treatments had similar grain yields and they did not imply that overapplication of nutrients reduced yields. Please ensure that other people’s work is summarized correctly. It would be best to check that this type of error does not occur elsewhere in the document.

Figure 2: This diagram is very difficult to read. Perhaps stacking A and B on top of one another instead of having them side by side will allow room for the text to be enlarged and the bars to be able to be more easily distinguished.

Lines 242-267: I am not sure why yields are only discussed in the context of precipitation and not both precipitation and temperature. I am also not convinced that fitting quadratic or cubic models to the effect of rainfall in certain time periods to grain yield makes ecological sense. The low r2 values support my point.

Table 4. If the treatment by year interaction is significant for corn grain P removal, why are those values are not shown? If an interaction is significant, main effects cannot be interpreted because they rely on each other. I suggest removing the letters of significance from the table in that particular column, then perhaps add a figure with the interaction effects.

Figure 4. As stated earlier, just because you can fit a cubic or quadratic model to data, does not mean it is ecologically meaningful. Are there other papers that suggest that as grain P concentration increases, yields will increase, then decrease, and then begin to increase again? It does not make ecological sense so it’s hard to understand the reasoning to do it. Consider a linear model or if that is not significant the perhaps exclude this figure.

Lines 288-290: I am not sure why these citations are included and how they are relevant to the current study. The P concentrations in grain in the current study did not appear to be impacted over the long term by higher application rates, as seen in Table 5. Most of the values were statistically the same for LMN and LMP and then also for SMP and SMN. So the current study disagrees with [58] (although please review this citation to ensure this is what was stated). The study in [59] evaluated nutrient uptake of barley, which may not be relevant here.

Lines 291-296: Please edit this section based on my comments for Figure 4.

Lines 327-328: Were soil samples taken to these lower depths prior to the beginning of the study to truly understand changes?

Line 336: Were fertilizers and manure applied at similar loading rates? Is this referring to total P applied or available P applied? Please clarify.

Lines 345-346: Are the levels of Olsen STP seen in this study expected to be high enough for P leaching to occur? Please discuss whether or not this is the case.

Line 347: Was bulk density impacted by treatments? If so, how did that influence SMS calculations? This was not discussed.

Line 356-358: Would a test for available P measure P stored in organic matter? It is unlikely. Perhaps if total P in the soil had been measured, this could be argued, but it does not really make sense in the case of this study.

Line 369: Again, it would be nice to know whether bulk density influenced SMS and therefore PSAC.

Lines 397-383: It is unclear how this adds to the discussion. According to what is written, liquid swine manure was compared to manure from other livestock, so how does this apply to liquid versus solid swine manure?

Line 386: How does the current study compare with a no-till study? I am having a hard time making this link.

Line 387-388: Again, not sure how application of swine manure to a pasture compares with the current study which was annual crops. Please make the connection or remove this.

Line 388-389: I am not sure how the current study implies there were no pollution effects on soil. This does not seem to be something that was evaluated. I did not see any discussion of STP levels and how they impact the environment. Is there a Olsen STP level in which leaching is likely to occur?

Lines 389-391: I am having a hard time following this logic. How does a coefficient for P solubility relate to a P availability coefficient? Also, it seems as if the PSAC is only calculated based on long-term studies which may not be relevant for most other studies that are only a few years long.

Line 395: The N-based strategy and the P-based strategy for solid manure had the same yield results. This sentence implies that was not the case.

Line 396-397: It was never discussed why this might have occurred. Likely due to immobilization of N from the bedding in the manure.

Line 398-400: Increased Olsen-P compared to what? There were no initial soil samples taken to understand if this was true over time. Perhaps the authors meant compared with the control?

Line 404: No evidence was given that PSAC is related to water quality, so this does not seem like a logical conclusion.

Line 406: Again, there was no relevant discussion of STP levels and water quality, so I'm not sure that this conclusion is supported by the data.

Reviewer 2 Report

The introduction should talk about N as well. Some English words need to be fixed
